# Response of Soil Environment and Microbial Community Structure to Different Ratios of Long-Term Straw Return and Nitrogen Fertilizer in Wheat–Maize System

**Man Yu [1], Qingxia Wang [1], Yao Su [1], Hui Xi [1], Yuying Qiao [1], Zhanlin Guo [2], Yunlong Wang [1] and Alin Shen [1,2,\***

[1]   Institute of Environment, Resource, Soil and Fertilizer, Zhejiang Academy of Agricultural Sciences, Hangzhou 310021, China
[2]   Institute of Plant nutrient, Environment and Resource, Henan Academy of Agricultural Sciences, Zhengzhou 450002, China
\*   Correspondence: shenalin@zaas.ac.cn; Tel.: +86-571-86409728

**Abstract:** To determine the reasonable rate of straw return and nitrogen (N) fertilizer use which may maintain soil ecosystem health, we analyzed their soil microbial biomass and composition in a 10-year field experiment with different rates of straw return (50%, 100%) and N fertilizer (270, 360, 450, 540 kg N ha$^{-1}$ yr$^{-1}$) by phospholipid fatty acid (PLFA) analysis and high-throughput sequencing. A rate of 50% straw return combined with 450 or 540 kg N ha$^{-1}$ yr$^{-1}$ effectively increased the soil available nutrient contents mainly for total nitrogen, available potassium, and available phosphorus. Total PLFAs indicated that straw return combined with N fertilizer promoted soil microbial growth and increased biomass. A rate of 100% straw return with 450 kg N ha$^{-1}$ yr$^{-1}$ was not conducive to the stability of the soil ecosystem according to the ratio of fungi to bacteria (F:B). The similar rate of straw returning and the similar level of nitrogen fertilizer application will be divided into the same cluster using a heatmap analysis. Some saprophytic fungi or pathogens became the dominant fungi genera, such as *Gibberella*, *Sarocladium*, *Pseudallescheria*, and *Mycosphaerella*, in the treatments with 100% straw returning combining higher N fertilizer (>450 kg ha$^{-1}$ yr$^{-1}$ yr$^{-1}$ added). The relative abundances of some heavy metal-tolerant bacteria, such as those in Proteobacteria and Chlorobi, increased in the soils in the 100% straw return treatments. Therefore, the combined application of 100% straw returning and higher N fertilizer (>450 kg ha$^{-1}$ yr$^{-1}$) added long-term was not appropriate for soil health, which will lead to the risk of disease and pollution in soil.

**Keywords:** straw return; nitrogen fertilizer; phospholipid fatty acids; high-throughput sequencing; soil health

## 1. Introduction

The full use of straw is highly important in stabilizing agricultural ecological balance, alleviating resource constraints, and reducing environmental pressure. The incorporation of crop straw into soil has become a common agricultural practice in China [1]. The addition of wheat straw contributed to increased plant nitrogen (N) uptake through the alternation between N immobilization early in the season and N re-mineralization late in the season [2], whereas the N plant bioavailability may decrease in the conditions of using straw without additional nitrogen fertilization as the rate of the organic nitrogen which can be taken up by the crop [3]. Moreover, excessive return of straw is always a challenge to crop production and soil quality maintenance. For example, straw can interfere with the planting of the crop, introduce pathogens or pests, and fix nutrients in soil and fertilizer [4]. Therefore, to improve the sustainable production of crops and maintain ecosystem health, a reasonable rate of straw return needs to be determined.

Compared with the treatment of single straw application, straw return with N fertilizer can substantially increase soil carbon (C) sequestration and reduce the intensity

of greenhouse gas emissions, thereby slowing down the greenhouse effect [5,6]. Under the condition of equal N inputs, the cumulative $N_2O$ emissions from 5000 kg ha$^{-1}$ and 10,000 kg ha$^{-1}$ straw return decreased by 23.1% and 33.5%, respectively, compared with the no-straw treatment [7]. However, excessive N application can lead to a variety of environmental problems, such as nitrate contamination of groundwater, soil acidification, and N runoff losses [8,9]. Under the same condition of straw returning, the accumulation of nitrate was higher in the treatment of 240 kg N ha$^{-1}$ at a depth of 200 cm, and the content of nitrate in the upper soil layer was lower, which indicated that nitrate is highly susceptible to leaching [10]. Liu et al. [11] showed that the optimal N application rate under different straw mulching conditions was between 117 and 178 kg N ha$^{-1}$. Therefore, it is necessary to further compare the effects of different N application rates on the soil environment.

Shifts in bacterial community structure or diversity and associated physiological responses can be used as indicators of these perturbations or disturbances in agroecosystems [12]. Soil physical and chemical properties, straw quality and return rate, and climate change greatly affect soil microbial communities [13]. In the short term, straw return can have either a major or a minor beneficial effect on soil microbial communities [14]. Straw return substantially increases the activities of invertase, protease, urease, and dehydrogenase, and also increases the biomass of Gram-positive bacteria and fungi [15]. After long-term straw return, the diversity of soil microbial communities increases, with the proportion of fungi decreasing and the proportions of bacteria and actinomycetes increasing [16]. Under the condition of straw returning, excessive application of N fertilizer not only inhibits the growth and reproduction of soil microorganisms but also inhibits microbial enzymatic activity [17]. To date, many studies have focused on the effects of different tillage methods, straw return methods, land types, and land use patterns on microbial communities [18–20], but the effects of different straw return rates and N application levels on microbial communities remain unclear.

Therefore, it is important to study the changes in soil microbial communities after long-term straw return and N fertilizer application in the field, focusing on the effects on soil health and ecology to develop more sustainable and beneficial straw return management in a rotation system.

## 2. Materials and Methods

### 2.1. Site Description and Experimental Setup

A 10-year field experiment was established in 2006 in Junxian County, Henan Province, in north-central China (35°67′ N, 114°54′ E). The soil type is cinnamon, developed through the process of viscosity and calcification. The site has a typical continental monsoon climate with mean annual temperature that ranges from 12.1 °C to 15.7 °C, mean annual precipitation that ranges from 532 to 1380 mm, sunshine duration that ranges from 1848 to 2488 h, and a frost-free period that ranges from 189 to 240 d.

The experiment was conducted in a winter wheat–summer maize rotation system using a split-plot design with three replicates per treatment. The two main-plot treatments were different rates of wheat and maize straw return, and the two subplot treatments were different rates of N fertilizer. The plot size was 55 m$^2$ (11 m × 5 m). The control plots did not receive fertilizer or straw. Straw return was at the rates of 3750 kg ha$^{-1}$ (S1, equal to approximately 50% of the crop yield) and 7500 kg ha$^{-1}$ (S2, equal to approximately 100% of the crop yield) under combined P and K fertilization at 210 kg $P_2O_5$ ha$^{-1}$ yr$^{-1}$ and 240 kg $K_2O$ ha$^{-1}$ yr$^{-1}$, respectively. All wheat and maize straw (except crop stubble) was removed from the plots before sowing of maize or wheat, and the straw was chopped to about 5–10 cm. The levels of N fertilizer applied were 270 (N1), 360 (N2), 450 (N3), and 540 (N4) kg N ha$^{-1}$ yr$^{-1}$. During the wheat season, N fertilizer was applied at sowing (60%) and jointing (40%); P (120 kg $P_2O_5$ ha$^{-1}$) and K (120 kg $K_2O$ ha$^{-1}$) fertilizer were applied before planting crops. During the maize season, N fertilizer was applied at sowing (40%) and flare opening (60%); P (90 kg $P_2O_5$ ha$^{-1}$) and K (120 kg $K_2O$ ha$^{-1}$) fertilizers were applied before planting crops. The fertilizers were broadcast, and the chemical

nitrogen, phosphorus, and potassium fertilizers were urea (46%), calcium superphosphate ($P_2O_5$ 12%), and potassium chloride ($K_2O$ 60%). Therefore, at the straw return rate of 3750 kg ha$^{-1}$, the treatments were S1N1, S1N2, S1N3, and S1N4, whereas at the straw return rate of 7500 kg ha$^{-1}$, the treatments were S2N1, S2N2, S2N3, and S2N4. Before the next crop-sowing season, the crushed straw would be directly returned to the soil then turned over to make it incorporate into the upper 20 cm of the soil.

### 2.2. Soil Sampling and Processing

Soil samples were collected 3 days after the wheat harvest prior to the straw returning and fertilizing in May 2016. Five soil cores (0 to 20 cm depth) were collected in each plot and mixed as a composite sample. Stones and plants were removed, and the soils were divided into two subsamples: one was freeze-dried and stored at $-70$ °C for microbial phospholipid fatty acid (PLFA) and DNA analyses; the other was air-dried and sieved through a 0.15 mm mesh for physicochemical analyses. After soil homogenization had taken place, PLFA analyses, DNA extraction, and soil physicochemical analyses were conducted, with three analytical repetitions for each sample site.

### 2.3. Soil Physicochemical Analyses

The physicochemical properties of soil samples were determined via traditional means. Briefly, soil pH was determined in a 1:2.5 (soil:water) solution by using a pH meter (Mettler Toledo, Columbus, OH, USA). The available nitrogen (AN) was determined by 1 mol L$^{-1}$ NaOH alkaline hydrolysis diffusion in 40 °C for 24 h and determined by HCl titration [21]. Soil organic carbon (SOC) was determined by the $K_2Cr_2O_7$-volumetric method; total nitrogen (TN) was determined by Kjeldahl's method (K9860, Haineng, Jinan, China); available phosphorus (AP) was extracted using 0.5 mol L$^{-1}$ NaHCO$_3$ and determined by the molybdenum blue colorimetric method (Shimadzu UV-2550, Kyoto, Japan); available potassium (AK) was extracted using 1 mol L$^{-1}$ NH$_4$OA and was determined by flame photometer (Blended Wing Body, BWB-XP, Cambridge, England); and electrical conductivity (EC) was measured in 1:5 soil/water suspensions and was determined by conductivity meter [22]. The soil carbon-to-nitrogen ratio (C/N) was calculated using SOC and TN.

Heavy metal analyses were conducted by the National Research Center for Geoanalysis (Beijing, China) according to the Soil Environmental Quality Standards in China (GB 15618-2018) (State Environmental Protection Administration of China, 2018). The total concentrations of heavy metals (including copper (Cu), lead (Pb), zinc (Zn), cadmium (Cd), nickel (Ni), and chromium (Cr)) in soils were extracted by a digestion mixture of HF, HNO$_3$, and HClO$_4$ and were analyzed by ICP-MS (Analytik jena, Plasma Quant MS elite, Germany). After solvent extraction-antiextraction, the total concentrations of mercury (Hg) and arsenic (As) in soils were determined by atomic fluorescence spectrophotometer (AFS-9230, China). Standard reference materials, NSA-5, obtained from the Institute of Geophysical and Geochemical Exploration, Chinese Academy of Geological Sciences, were analyzed as part of the quality assurance and quality control procedures. Good agreement was achieved between the data obtained in the present work and the certified values. The analysis of the samples, including soil samples and blanks, was performed in triplicate, and the standard deviation was within 5%.

### 2.4. Phospholipid Fatty Acid Analysis

Triplicate subsamples of ~3.0 g of freeze-dried soil from each treatment were used to extract PLFAs using the method described by Yu et al. (2009) [23]. The phospholipid ester-linked fatty acid methyl esters were dissolved in hexane for further GC (gas chromatography) (Agilent Technologies 7890A, Santa Clara, CA, USA) analysis and quantitated by Sherlock 6.1 (Microbial Identification System). Nonadecanoic acid methyl ester (19:0) was used as an internal standard. Total PLFA concentration is expressed in units of nmol·g$^{-1}$.

### 2.5. DNA Sequencing Analysis

DNA extraction was performed on each composite of ~1.0 g of soil sample using Mobio Power Soil DNA Isolation Kit (Cat.#12888-50, Qiagen, Germantown, MD, USA) according to the instruction manual. The extracted DNA was quantified using a Nanodrop ND-1000 spectrophotometer (Thermo Fisher Scientific, Waltham, MA, USA). The extracted DNA was analyzed using Illumina MiSeq sequencing (Illumina, San Diego, CA, USA). Polymerase chain reaction (PCR) amplification was conducted using the primer set of 27F (5′-ACACTGACGACATGGTT-3′) [24] and 533R (5′-TTACCGCGGCTGCTGGCAC-3′) [19] for the V1–V3 region of the bacterial 16S rRNA gene and the primer set of ITS1 (5′-CTTGGTCATTTAGAGGAAGA GAA-3′) and ITS2 (5′-GCTGCGTTCTTCATCG ATGC-3′) for the ITS region of fungi [25]. The PCR reaction was performed in a 20 μL mixture containing 4 μL of 5 × FastPfu buffer, 2 μL of 2.5 mmol $L^{-1}$ dNTPs, 0.8 μL of each primer (5 μmol $L^{-1}$), 0.4 μL of FastPfu polymerase, 10 ng of template DNA, and dd$H_2O$ to bring the mixture to volume. The PCR conditions were as follow: 95 °C for 3 min, followed by 27 cycles at 95 °C for 30 s, 55 °C for 30 s, 72 °C for 45 s, and a final extension at 72 °C for 10 min (GeneAmp® 9700, ABI, Foster City, CA, USA). Amplicons were extracted and purified using an AxyPrep DNA Gel Extraction Kit (Axygen Bioscience, Union City, CA, USA) according to the manufacturer's instructions. Purified amplicons were pooled in equimolar ratios and paired-end sequenced on an Illumina MiSeq platform according to standard protocols. The raw reads were quality-filtered by Qiime (version 1.9.1, http://qiime.org/install/index.html, accessed on 5 January 2023) and merged by FLASH (version 1.2.11, https://ccb.jhu.edu/software/FLASH/index. shtml, accessed on 5 January 2023) according to the following criteria: (1) reads were truncated at any site receiving an average quality score < 20 over a 50-bp sliding window; (2) sequences whose overlap was longer than 10 bp were merged according to their overlap, with mismatch no more than 2 bp; and (3) sequences of each sample were separated according to barcodes (exactly matching) and primers (allowing two nucleotide mismatches), and reads containing ambiguous bases were removed. Operational taxonomic units (OTUs) were clustered according to 97% similarity cutoff using UPARSE (version 7.1; http://drive5.com/uparse/, accessed on 5 January 2023) with a novel greedy algorithm that performed chimera filtering and OTU clustering simultaneously. The taxonomy of each 16S rRNAor ITS gene sequence was analyzed by RDP Classifier algorithm (http://rdp.cme.msu.edu/, accessed on 5 January 2023) against the databases of SILVA128 (http://www.arb-silva.de, accessed on 5 January 2023) for the ITS region, using a confidence threshold of 70%. Sequencing reads were deposited in the Sequence Read Archive at the NCBI under the accession number of SRP268603 under the Project ID of PRJNA 641490.

### 2.6. Statistical Analyses

All statistical analyses were performed with SPSS 19.0 (SPSS Inc., Chicago, IL, USA). Soil physicochemical properties and heavy metal contents were subjected to a two-way (straw return × nitrogen fertilization, S × N) analysis of variance (ANOVA) in a split-plot arrangement followed by the least significant difference test (LSD, $p < 0.05$). The differences in the composition of soil microbial communities were investigated by principal component analysis (PCA) using CANOCO program for Windows 4.5—software (CANOCO, Microcomputer Power Inc., Ithaca, NY, USA). The heatmap and correlation coefficient are analyzed using the packages "pheatmap" (https://cran.r-project.org/web/packages/pheatmap/index.html, accessed on 5 January 2023) and "corrplot" (https://cran.r-proje ct.org/web/packages/corrplot/index.html, accessed on 5 January 2023) based on statistical software R version 3.6.1 (https://www.r-proje ct.org/, accessed on 5 January 2023). Spearman correlation analysis was used to test the statistical significance of the relationship between the variation in bacterial and fungal community structure with the soil physiochemical properties.

## 3. Results

### 3.1. Soil Physicochemical Properties and Heavy Metal Contents

The soil physicochemical properties contents in the different treatments are shown in Figure 1 and Tables S1–S3. Except for S1N1 soil (8.57 g kg$^{-1}$), the SOC level in all other treatments was higher than that in the control (8.92 g kg$^{-1}$). With the increase in N fertilizer application, the soil SOC content at the same rate of straw return treatments increased significantly. The increase in SOC at 50% straw return with 450 kg N ha$^{-1}$ yr$^{-1}$ and 100% straw return with 540 kg N ha$^{-1}$ yr$^{-1}$ was particularly notable. Soil TN in S1N4 (1.27 g kg$^{-1}$) was higher than that in the other treatments, in which TN ranged from 0.85 (control) to 1.06 (S2N4) g kg$^{-1}$, while soil AN in S1N4 (27.0 mg kg$^{-1}$) was mostly lower than that in the other treatments. This suggested that the rate of organic nitrogen in soil TN was highest in S1N4. The changes of treatments on soil C/N ratios were similar to those on SOC, which were affected by the levels of N fertilizer input and the interaction of straw and N fertilizer application. Compared with the control, straw and N fertilizer application significantly increased the AK and AP content in each treatment ($p < 0.001$), but there was little variation in AK content between the treatment with different levels of straw return. The highest soil AP content was in S1N2 (22.24 mg kg$^{-1}$), and the soil AP contents in the 50% straw return treatments were higher than that in the 100% straw return treatments. The control had the highest soil pH of 8.90, whereas in the treatments, pH ranged from 8.08 to 8.29. Although there was a little decrease in pH compared to the control, no significant effect on it was found from straw and N fertilizer application.

The heavy metal contents in the different treatments are shown in Figure S1 and Table S4. In the control, the mean contents of heavy metals were the following: Pb, 0.05 mg kg$^{-1}$; Zn, 4.77 mg kg$^{-1}$; Cr, 60.94 mg kg$^{-1}$; Cu, 22.63 mg kg$^{-1}$; Ni, 78.15 mg kg$^{-1}$; and As, 0.01 mg kg$^{-1}$. The average contents of Cd (0.15 µg kg$^{-1}$) and Hg (0.05 µg kg$^{-1}$) in the control soil were much lower than those of the other heavy metals. The thresholds of heavy metals according to the Soil Environmental Quality Standards in China (GB15618-2018) are the following: Pb, 170.00 mg kg$^{-1}$; Cd, 0.60 mg kg$^{-1}$; Zn, 300.00 mg kg$^{-1}$; Cr, 250.00 mg kg$^{-1}$; Cu, 100.00 mg kg$^{-1}$; Ni, 190.00 mg kg$^{-1}$; Hg, 3.40 mg kg$^{-1}$; and As, 25.00 mg kg$^{-1}$. The mean contents of all the heavy metals in soils did not exceed the thresholds. Compared with the control soil, the contents of Cd and Zn in the treatments' soil were higher, which may indicate that straw return and fertilizer contribute to the accumulation of Cd and Zn in soil.

### 3.2. Phospholipid Fatty Acid Analysis of Soil Microbial Communities

A total of 19 monomeric PLFAs were detected in all treatments. The total PLFA content and the distribution among major taxa of the microbial communities in each treatment are shown in Figure 2a. Compared with the control at 96.18 nmol g$_{soil}$$^{-1}$, the PLFA content increased significantly in the soils treated with different amounts of straw return except for S2N4 (97.13 nmol g$_{soil}$$^{-1}$), ranging from 131.23 (S1N3) to 511.48 (S2N3) nmol g$_{soil}$$^{-1}$. The biomass was highest in S1N1 and S2N3 soils. The highest fungi: bacteria (F:B) ratio was in S1N4, whereas the lowest ratios were in S2N3 and the control (Figure 2b).

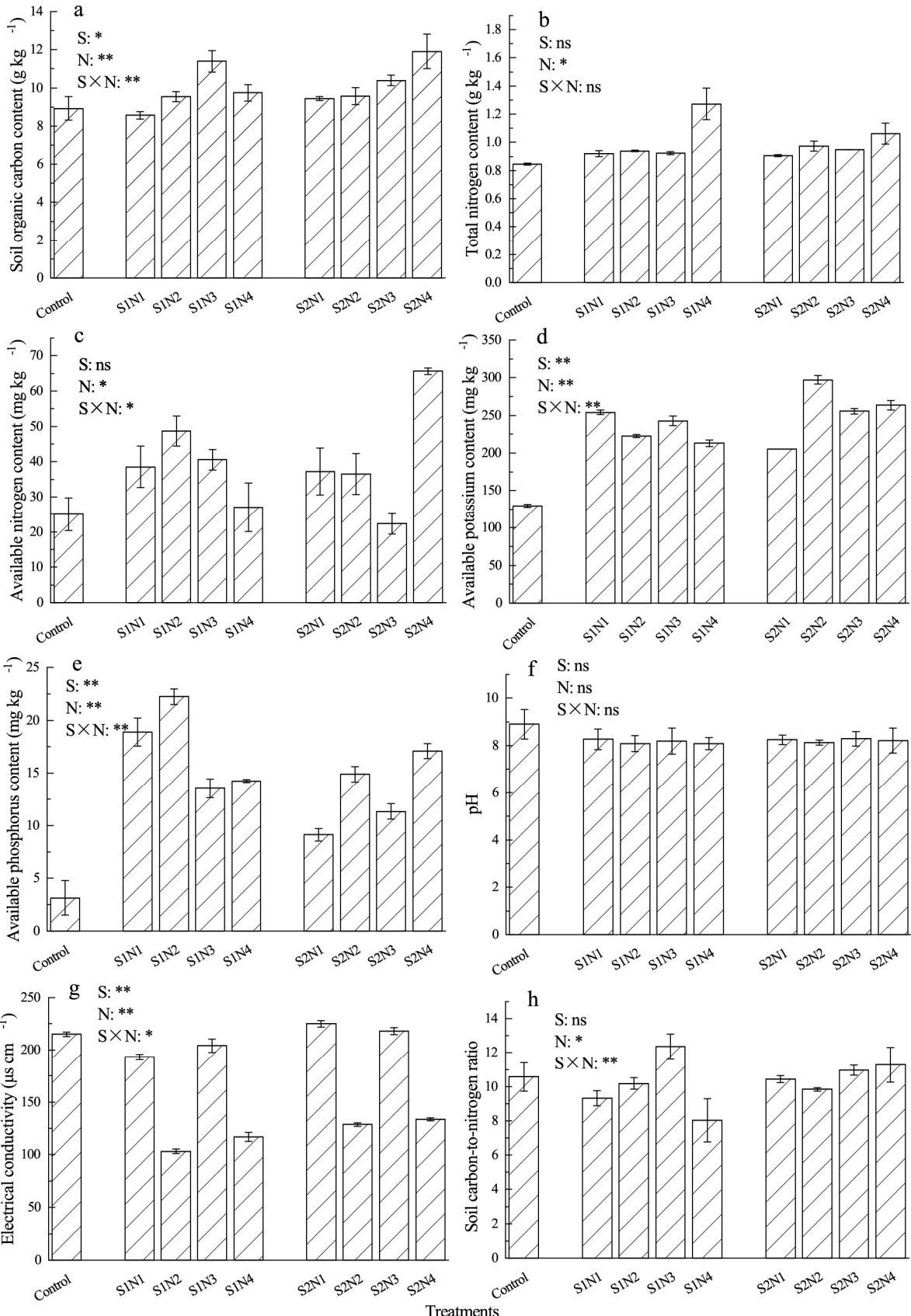

**Figure 1.** Soil physicochemical properties contents ((**a**) soil organic carbon content (g.kg$^{-1}$), (**b**) total nitrogen content (g.kg$^{-1}$), (**c**) available nitrogen content (mg.kg$^{-1}$), (**d**) available potassium content (mg.kg$^{-1}$), (**e**) available phosphonus content (mg.kg$^{-1}$), (**f**) pH, (**g**) electrical conductivity, (**h**) soil

carbon-to-nitrogen ratio) after long-term (10 yr) straw return and N fertilizer treatments in a wheat–corn rotation in Henan Province, China. Notes: Control—no fertilizer or straw. The straw return rates were 3750 (S1) and 7500 (S2) kg ha$^{-1}$ yr$^{-1}$. The N fertilizer applied levels were 270 (N1), 360 (N2), 450 (N3), and 540 (N4) kg N ha$^{-1}$ yr$^{-1}$. Error bar represents standard error. *, **, and ns represent $p < 0.05$, $p < 0.001$, and no significant differences, respectively.

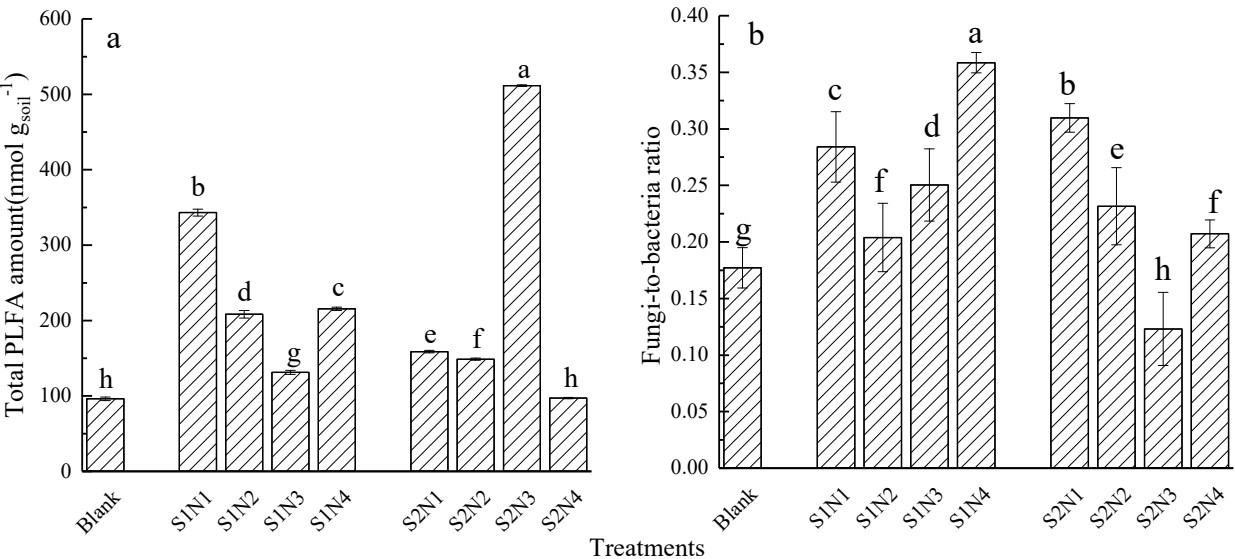

**Figure 2.** (**a**) Total phospholipid fatty acid (PLFA) content (nmol g$_{soil}$$^{-1}$). (**b**) Fungi-to-bacteria ratio based on PLFAs in soils with different rates of straw return and N fertilizer in a long-term (10 yr) wheat–corn rotation. Notes: Control—no fertilizer or straw. The straw return rates were 3750 (S1) and 7500 (S2) kg ha$^{-1}$. The N fertilizer applied levels were 270 (N1), 360 (N2), 450 (N3), and 540 (N4) kg N ha$^{-1}$ yr$^{-1}$. Error bar represents standard error. Small letters a–h represent significant differences.

### 3.3. Taxonomic Composition Based on Miseq Sequencing

To further analyze the changes in taxonomic composition of soil microbial communities in response to straw return and N fertilization, the composition of bacterial and fungal communities was characterized using Miseq sequencing. Heatmaps are drawn and presented in Figure 3 based on the top 40 bacteria and fungi at genus level. Dendrograms based on the variation matrices of the datasets show the overall grouping structures of the parts of the respective compositions according to their codependence. The dendrogram of the bacterial dataset presents two clear groupings of codependent components (Figure 3a). The first group comprises S1N4 alone, which is clearly the most independent element in the bacterial dataset. The other group comprises S2N3−S1N3, control−S1N1, S2N1−S2N2, and S1N2−S2N4. The dendrogram of the fungal dataset also presents two groups of codependent components (Figure 3b), the first formed of S1N3−S2N4, and the second formed of S1N2−S2N1−S2N2, control−S1N1, and S2N3−S1N4. According to the heatmap analysis, the similar rate of straw returning and the similar level of nitrogen fertilizer application will be divided into the same cluster.

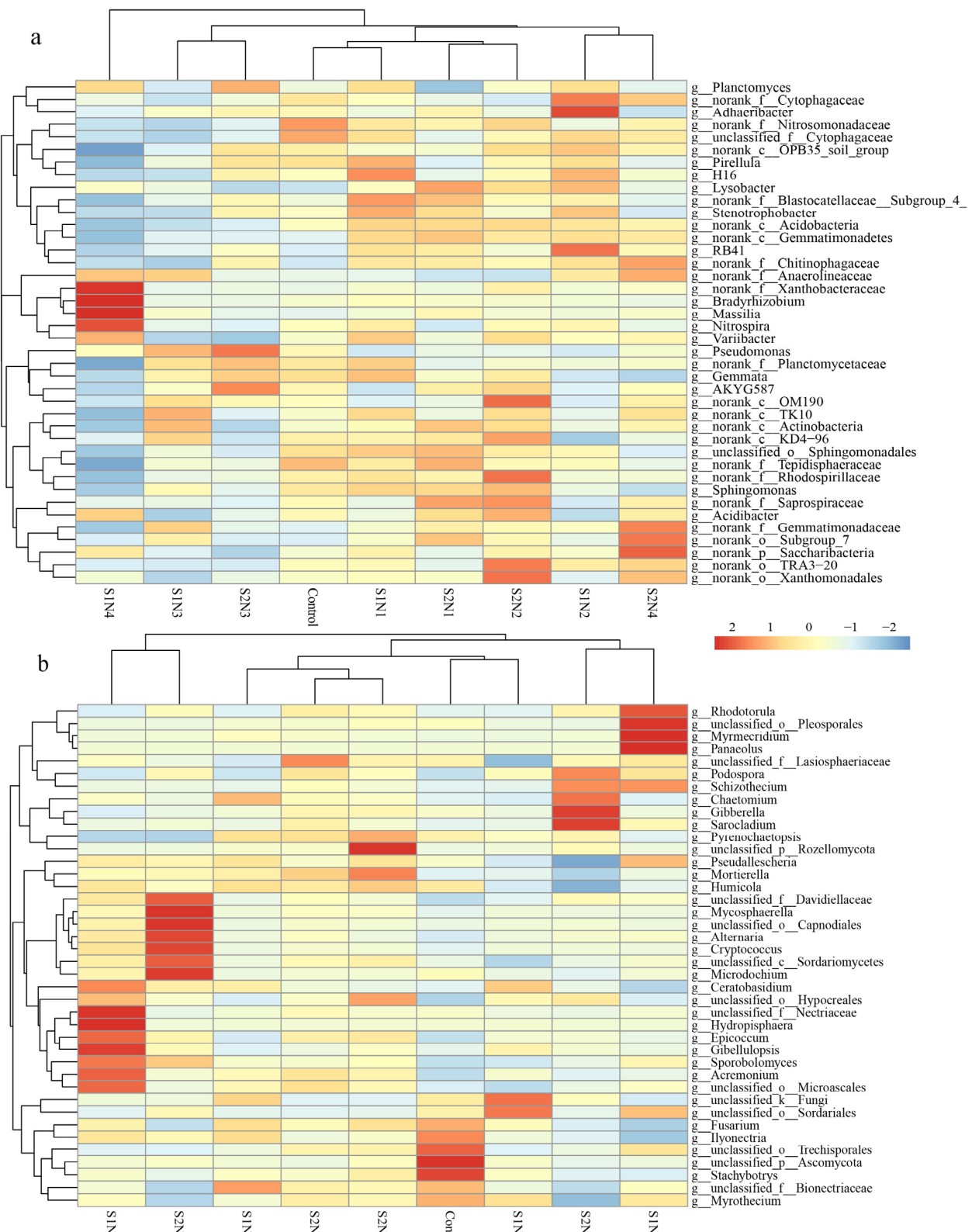

**Figure 3.** Heatmap of correlation on genus combined with UPGMA cluster analysis between the relative abundance of series of bacteria (**a**) and fungi (**b**) genus and samples, including the relative richness (n = 3) of the 40 dominant microbial genera. Note: Horizontal ordinates represent soil samples; vertical ordinates represent bacterial community abundance information. Control—no fertilizer or straw. The straw return rates were 3750 (S1) and 7500 (S2) kg ha$^{-1}$ yr$^{-1}$. The N fertilizer applied levels were 270 (N1), 360 (N2), 450 (N3), and 540 (N4) kg N ha$^{-1}$ yr$^{-1}$.

The cluster heatmap of 40 dominant genera of bacteria or fungi in different treatment soils is shown in Figure 3. The redder the rectangle color is, the larger the relative abundance proportion of this genus; conversely, the bluer the rectangle color is, the smaller the relative abundance proportion of this genus. As shown in Figure 3a, the main bacteria phyla in this field were *Proteobacteria*, *Verrucomicrobia*, and *Planctomycetes*. The genera of *Bradyrhizobium*, *Massilia*, the families of *Xanthobacteraceae*, *Planctomycetaceae*, *Tepidisphaeraceae*, and the class of *OPB35* are dominant in S1N4; the genus of *Adhaeribacter* is the main bacteria in S1N2.

In Figure 3b, in treatments with 50% straw returning, the genus of *Hydropisphaera* and the family of *Nectriaceae* are dominant in S1N3, and the genera of *Myrmecridium* and *Panaeolus* and the order of *Pleosporales* are the main fungi in S1N4. While in treatments with 100% straw returning, the genera of *Gibberella*, *Sarocladium*, and *Pseudallescheria* are dominant in S2N3, and the genera of *Mycosphaerella* and *Microdochium* and the order of *Capnodiales* are the main fungi in S2N4. There are no obvious major fungi genera in treatments with lower N fertilizer in S1N1-N2 and S2N1-N2, which is similar to the control.

### 3.4. Multivariate Correlation Analysis between Microbial Communities and Soil Environmental Variables

Differences in straw return treatments may alter the chemical properties of soils, and the changes are often closely related to changes in soil microbial communities [26]. Spearman correlation coefficients were calculated to evaluate the relationship between microbial communities and environmental factors on the basis of phylum-level information from all soil samples, as shown in Figures 4 and 5. The environmental variables, including soil physicochemical parameters (SOC, TN, AN, AK, AP, C/N, pH, and EC) and heavy metals (Cu, Cd, Pb, Cr, As, Hg, Ni, and Zn), were influence factors that were significantly correlated with the taxa of the microbial communities.

Among the heavy metals, in treatments with 50% straw return, it showed that Cd was significantly negatively correlated with *Bacteroidetes* (r = −0.959, $p$ = 0.041) and *Fibrobacteres* (r = −0.990, $p$ = 0.010). The relative abundance of *Firmicutes* was significantly negatively correlated with soil C/N (r = −0.952, $p$ = 0.045) and Cr (r = −0.961, $p$ = 0.039). In addition, the relative abundance of *Chloroflexi*, *Gemmatimonadetes*, *Chlorobi*, *Elusimicrobia*, and unclassified bacteria was significantly positively correlated with soil AP (r = 0.984, $p$ = 0.016), AN (r = 0.998, $p$ = 0.033), As (r = 0.975, $p$ = 0.025), Pb (r = 0.965, $p$ = 0.015), and EC (r = 0.977, $p$ = 0.023). The relative abundance of *Planctomycetes*, *Nitrospirae*, and *Verrucomicrobia* showed a significant negative correlation with soil TN (r = −0.998, $p$ = 0.018), Pb (r = −0.965, $p$ = 0.035), and SOC (r = −0.952, $p$ = 0.048) (Figure 4a).

In the 100% straw return treatments, *Bacteroidetes* was significantly correlated with >2 heavy metals, and more bacterial groups were significantly positively correlated with heavy metals than those in the 50% straw return treatments. It showed that the relative abundance of *Bacteroidetes* was significantly positively correlated with As (r = 0.956, $p$ = 0.044), Cr (r = 0.958, $p$ = 0.042), and Pb (r = 0.979, $p$ = 0.021). The relative abundance of *Gemmatimonadetes* and *Verrucomicrobia* was significantly positively correlated with soil TN (r = 0.956, 0.956, $p$ = 0.044, 0.044) and AP (r = 0.969, 0.995, $p$ = 0.031, 0.039). The soil AN was significantly positively correlated with *Armatimonadetes* (r = 0.974, $p$ = 0.026), while it was negatively correlated with *Cyanobacteria* (r = −0.974, $p$ = 0.026). Hg was significantly positively correlated with *Chlorobi* (r = 1.000, $p$ < 0.001) and others (r = 1.000, $p$ < 0.001). The relative abundance of *Proteobacteria* was significantly positively correlated with soil pH (r = 0.994, $p$ = 0.006) and Ni (r = 0.985, $p$ = 0.015). The relative abundance of *Candidate_division* was significantly positively correlated with Zn (r = 0.992, $p$ = 0.008). In addition, the relative abundance of *Acidobacteria*, *Planctomycetes*, and unclassified bacteria was significantly negatively correlated with soil EC (r = −0.951, $p$ = 0.049), SOC (r = −0.997, $p$ = 0.048), and TN (r = −0.965, $p$ = 0.035) (Figure 4b).

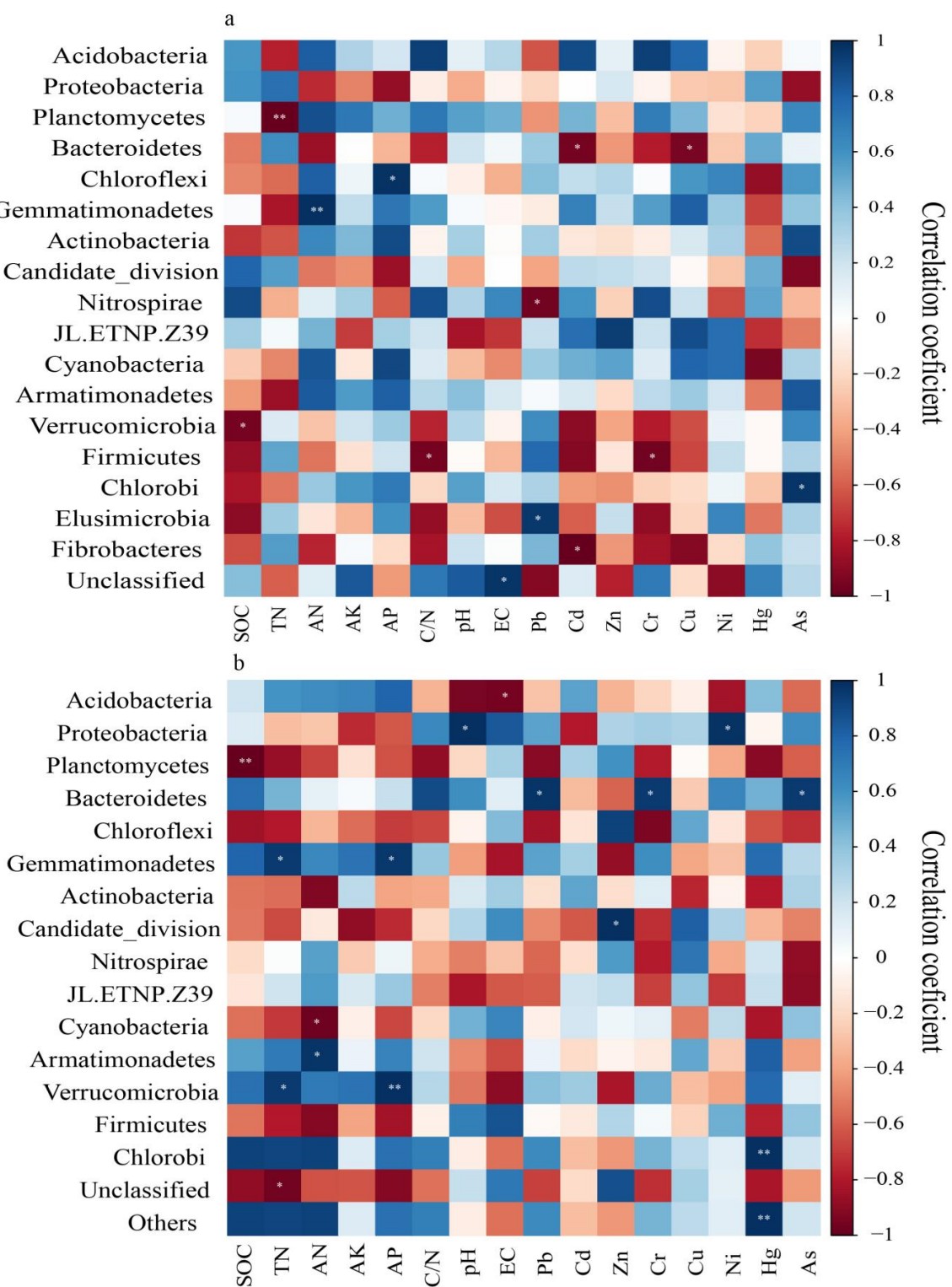

**Figure 4.** Correlation diagrams of positive (blue) and negative (red) correlations between soil physicochemical variables and heavy metal contents and taxa of bacterial communities in treatments receiving (**a**) 50% straw return and (**b**) 100% straw return in a long-term (10 yr) wheat–corn rotation. * indicates $p < 0.05$; ** indicates $p < 0.01$. Notes: TOC—total organic carbon; TN—total nitrogen; AN—available nitrogen; AK—available potassium; AP—available phosphorus; C/N—carbon-to-nitrogen ratio; EC—electrical conductivity; Pb—lead; Cd—cadmium; Zn—zinc; Cr—chromium; Cu—copper; Ni—nickel; Hg—mercury; As—arsenic.

In the 50% straw return treatments, the relative abundance of *Ascomycotawas* significantly negatively correlated with C/N (r = −0.983, *p* = 0.017) and Cr (r = −0.984, *p* = 0.016). The relative abundance of *Basidiomycota* was significantly negatively correlated with Pb (r = −0.961, *p* = 0.039) (Figure 5a).

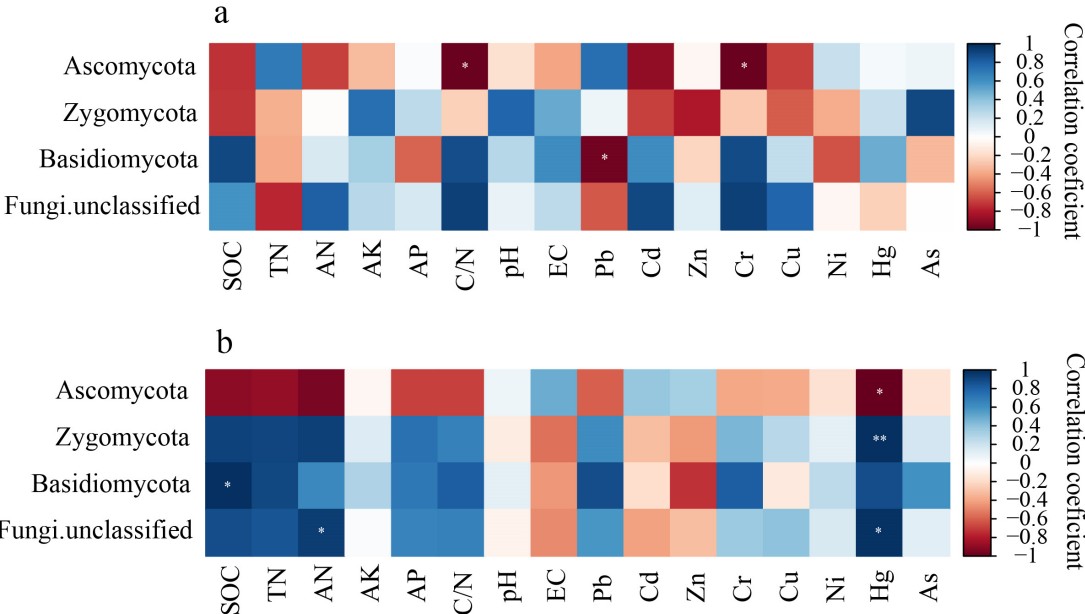

**Figure 5.** Correlation diagrams of positive (blue) and negative (red) correlations between soil physicochemical variables and heavy metal contents and taxa of fungal communities in treatments receiving (**a**) 50% straw return and (**b**) 100% straw return in a long-term (10 yr) wheat–corn rotation. * indicates *p* < 0.05; ** indicates *p* < 0.01. Notes: TOC—total organic carbon; TN—total nitrogen; AN—available nitrogen; AK—available potassium; AP—available phosphorus; C/N—carbon-to-nitrogen ratio; EC—electrical conductivity; Pb—lead; Cd—cadmium; Zn—zinc; Cr—chromium; Cu—copper; Ni—nickel; Hg—mercury; As—arsenic.

In the 100% straw return treatments, the relative abundance of *Ascomycota*, *Zygomycota*, and unclassified fungi was significantly positively or negatively correlated with Hg (r = −0.994, 1.000, 0.990; *p* = 0.006, 0.002, 0.030). The relative abundance of *Basidiomycota* and unclassified fungi was significantly positively correlated with SOC (r = 0.990, *p* = 0.010) and AN (r = 0.951, *p* = 0.005) (Figure 5b).

Overall, the bacterial community had a significantly relevance with SOC, TN, AN, AP, C/N, pH, EC, Cd, Pb, Cr, As, Hg, Ni, and Zn. The most relevant soil environmental variables to the fungal community were SOC, C/N, AN, Hg, Pb and Cr. It indicated that bacterial community was more sensitive and responsive to the environmental variables than the fungal community.

## 4. Discussion

### 4.1. Response of Soil Physicochemical Properties with Long-Term Straw Return and Nitrogen Fertilizer

Fertilization can promote microbial growth, increase biomass biosynthesis, and increase the effective microbial carbon conversion rate of straw; appropriate fertilizer management can therefore effectively inhibit reductions in soil C and N [27]. In addition, N fertilizer promotes crop growth and increases the secretion of root organic matter [28]. N application combined with straw returning can promote soil organic C sequestration and N retention [29]. However, excessive N application may inhibit root growth in deep soils, leading to nitrate leaching and pollution of aquifers [30]. The higher the N input, the higher the residual $NO_3^-$-N content in the top 200 cm of the soil profile, indicating greater potential for leaching of N [31]. The leaching of N can be an important contribution to soil N loss [32]. In addition, the pH of the soil in the treatments was higher than seven, which may result

a significant ammonia volatilization as another loss of N [8]. Soil AN was more closely correlated with the current soil, plant, and environmental parameters than soil TN. In the 50% and 100% straw return treatments, depending on the amount of fertilizer, soil AK and AP content also improved significantly, while the soil AN did not increase with the increase of N input. In the 50% straw return treatments with 450 kg N ha$^{-1}$ yr$^{-1}$, the TN was the highest with a lower AN, it indicated that the organic nitrogen in soil was higher in the S1N3.

*4.2. Response of Soil Microbial Communities with Long-Term Straw Return and Nitrogen Fertilizer*

Total PLFAs have been suggested as a measure of the total microbial biomass, and it has been shown to correlate well with the biomass measurements using fumigation extraction technique in our previous work [23]. In this experiment, the total tPLFA was used to suggest the changes in microbial biomass. The high-energy C and nutrient sources provided by straw support microbial growth and increase microbial biomass [33]. Total PLFAs indicated that straw return combined with N fertilizer promoted soil microbial growth and increased biomass. In general, fungi are tightly associated with soils with low nutrient availability and organic matter decomposition rates, whereas bacteria dominate in soils with relatively high nutrient availability and decomposition rates, and thus, an increase in F:B ratios could reflect reduced nutrient availability and slower growth rates in the soil [34]. F:B ratios can increase understanding of microbial communities and they can be used to predict changes in composition as well as responses to ecosystem disturbances [35,36]. The relative abundance of fungi is highly correlated with soil C/N ratios because high C/N ratios might inhibit the activities of soil microorganisms (especially those of bacteria) due to the lack of N in the soil [37]. The C storage in fungal biomass is expected to be higher and more persistent than that in bacterial biomass, and an increase in soil C storage leads to faster ecosystem recovery [25]. Therefore, the F:B ratio may reveal the stability of a soil ecosystem, with a high ratio representing high and sustainable stability [23]. In this study, the different rates of straw return and N fertilization had different effects on the structure of soil microbial communities. The low F:B ratios in the control and S2N3 treatments indicated that treatment with 100% straw and 450 kg N ha$^{-1}$ yr$^{-1}$ was not conducive to the stability of the soil ecosystem.

The combined use of inorganic fertilizers and organic materials provides both nutrients and appropriate C sources for the more active and symbiotic microbiota [38]. There are four bacterial taxa in the type range important for agricultural soils: *Actinobacteria* (35–39%), *Proteobacteria* (26–33%), *Acidobacteria* (11–13%), and *Verrucomicrobia* (3–6%). The other (less common) types are *Firmicutes* (3–6%), *Bacteroidetes* (2–3%), *Gemmatimonadetes* (1–2%), *Nitrospira* (1%), *Chloroflexi* (1–2%), and *Planctomycetes* (0–1%) [39]. *Proteobacteria* and *Bacteroidetes* are the main bacteria in soil after straw return [40,41]. Nitrogen content might induce directly or indirectly a shift in the predominant microbial community members, especially those of the Proteobacteria and Bacteroidetes phyla [42]. *Bacteroidetes* is a sensitive biological indicator of agricultural soil usage. Compared with the non-cultivated soil, the number of *Bacteroides* OTUs in the cultivated soil decreased [43]. The populations of some groups, such as *Nitrospirae, Gemmatimonadetes, Actinobacteria*, and *Proteobacteria*, are more sensitive to environmental changes [44]. Bacterial communities in S1N4 are categorically *Proteobacteria* (genera *Bradyrhizobium, Massilia*, and family *Xanthobacteraceae*), *Planctomycetes* (family *Planctomycetaceae, Tepidisphaeraceae*), and *Verrucomicrobia* (class of *OPB35*). *Bacteroidetes* (genus *Adhaeribacter*) is the main phylum of bacteria in S1N2. It is expected that *Proteobacteria* and *Bacteroidetes* will be the dominate phyla in soil after long-term straw and N fertilizer are added, which is consistent with the results of S1N4 and S1N2.

*Hydropisphaera, Nectriaceae, Myrmecridium, Panaeolus*, and *Pleosporales* are the major decomposers of soil fungal communities [45,46]. Members of *Basidiomycota* can degrade ligno-cellulosic organic matter more efficiently than other members of the fungal community [47], whereas the members of *Ascomycota* use recalcitrant lignin in the degradation of litter [48]. Most species within the genera *Gibberella, Sarocladium, Pseudallescheria, Mycosphaerella*, and

*Microdochium* are either saprophytic fungi in soil or pathogens of plants [49,50]. In particular, Gibberella is a representative pathogen causing fusarium head blight [51]. It is suggested that higher straw and N fertilizer added long-term will lead to the risk of soil disease with saprophytic fungi.

*4.3. The Correlation between Soil Microbial Communities and Environmental Variables*

The microbial communities contaminated by heavy metals appeared more diverse to a certain extent. The effects of heavy metals do not act simply on one species but on microbial populations in the sediment [52]. Several investigations have proven that both structural and functional diversity in polluted soil can greatly improve [53], and bacterial communities can exhibit structural and functional resilience to metals [54]. Although many elements are necessary in certain physiological processes, toxicity at high levels can inhibit the growth of microbes [55]. In the 100% straw return treatments, *Proteobacteria*, *Bacteroidetes*, *Candidate_division*, and *Chlorobi* were positively correlated with heavy metals (Figure 4b). Thus, the positive correlations may indicate greater microbial tolerance to heavy metals. *Proteobacteria*, *Acidobacteria*, *Gemmatimonadetes*, and *Planctomycetes* have strong heavy metal absorption and transfer abilities [56]. The variation in the response of *Proteobacteria* to heavy metals may be due to the complex metabolisms of bacteria in the phylum, which can use various types of organic matter as C, N, and energy sources [57]. This result suggested that in the 100% straw return treatments, the abundance of bacteria with heavy metal tolerance was promoted to resist the increasing concentration of heavy metals. In the fungal community, the same situation occurred. In the 100% straw return treatments, *Zygomycota*, *Basidiomycota*, and unclassified fungi were positively correlated with heavy metals (Figure 5b). The 100% straw return treatments had strong transfer and absorption capacity of heavy metals, which suggest a higher level of soil pollution [58].

**5. Conclusions**

The straw returning with N fertilizer increased the contents of most soil nutrient, especially in treatment at 50% straw return with 450 kg N ha$^{-1}$ yr$^{-1}$, mainly for higher SOC, TN, AK, and AP contents. The structure of soil microbial communities was also significantly affected by straw return and N fertilizer. Total PLFAs indicated that straw return combined with N fertilizer promoted soil microbial growth and increased biomass. The F:B ratio showed that 100% straw with 450 kg N ha$^{-1}$ yr$^{-1}$ was not conducive to the stability of the soil ecosystem. According to the heatmap analysis, the similar rate of straw returning and the similar level of nitrogen fertilizer application will be divided into the same cluster. However, some genera, including *Gibberella*, *Sarocladium*, *Pseudallescheria*, *Mycosphaerella*, and *Microdochium*, which are either saprophytic fungi in soil or pathogens of plants, became the dominant fungi genera in the treatments with 100% straw returning combining higher N fertilizer (>450 kg ha$^{-1}$ yr$^{-1}$) added. The fungal communities were significantly affected by heavy metals (Hg, Pb, and Cr), SOC, AN, and the C/N ratio. In addition, the relative abundance of some heavy metal-tolerant bacteria, such as those in *Proteobacteria* and *Chlorobi*, increased in the soils in the 100% straw return treatments. Thus, the results of this study revealed that long-term 100% straw returning combining higher N fertilizer (>450 kg ha$^{-1}$ yr$^{-1}$) might lead to the risk of soil disease and pollution, while 50% straw returning combined with 450 kg ha$^{-1}$yr$^{-1}$ might be more sustainable and beneficial to soil nutrient accumulation and microbial community health in wheat–maize rotation fields.

**Supplementary Materials:** The following supporting information can be downloaded at: https://www.mdpi.com/article/10.3390/su15031986/s1, Table S1: The effects of straw returning, N fertilizer and between-subjects straw*N on soil physicochemical variables. Table S2: Muliple comparisons of soil physicochemical variables with different rate of staw returning. Table S3: Muliple comparisons of soil physicochemical variables with different rate of N fertilizer application. Table S4: The effects of straw returning, N fertilizer and between-subjects straw*N on soil heavy metal. Figure S1: Soil heavy metal contents after long-term (10-yr) straw return and N fertilizer treatments in a wheat-corn rotation in Henan Province, China.

**Author Contributions:** Conceptualization, M.Y. and A.S.; methodology, M.Y.; software, Q.W.; validation, M.Y., Q.W. and Y.Q.; formal analysis, Q.W.; investigation, H.X. and Y.W.; resources, Z.G.; data curation, Y.S.; writing—original draft preparation, M.Y. and Q.W.; writing—review and editing, M.Y.; visualization, Y.Q.; supervision, A.S.; project administration, A.S.; funding acquisition, A.S. All authors have read and agreed to the published version of the manuscript.

**Funding:** This research was funded by China Modern Agriculture Research System, grant number CARS 03-28 and China agricultural Industry system construction special fund of wheat, grant number Yucaike (2015) 186-2060502.

**Institutional Review Board Statement:** Not applicable.

**Informed Consent Statement:** Not applicable.

**Data Availability Statement:** Not applicable.

**Conflicts of Interest:** The authors declare no conflict of interest.

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
