# Peer review of "Response of Soil Environment and Microbial Community Structure to Different Ratios of Long-Term Straw Return and Nitrogen Fertilizer in Wheat–Maize System"

_sustainability, doi:10.3390/su15031986_

Round 1

Reviewer 1 Report

  1. The keywords should not overlap with the title.
  2. Immobilization of N fertilizer can occur rapidly in soil (over a few weeks) but this N is released slowly. Some authors reported that 7% of the 15N-labelled organic nitrogen was taken up by the first cereal crop after the labeled fertilizer had been applied, 4% by the second crop, 2% by the third and fourth crops, and 55% of the immobilized 15N remained in the soil at the end of 4 years. Please consider the issues related to the reduction of N plant bioavailability in the conditions of using straw without additional nitrogen fertilization (L. 41-44).
  3. On what basis were the doses of mineral fertilization determined? (L.110-111, 113-114)
  4. The methodology does not mention a 10-year experiment, which is written in the abstract and the captions under the figures. 
  5. All methods used should be adequately described in the Materials and Methods section, e.g. method of available nitrogen determination. 
  6. Please correct editing errors, e.g. HClO4 (L. 158), significantdifferences (L. 259)
  7. What was the basis of this statement: L. 464-467, 560-565.

Author Response

  1. The keywords should not overlap with the title.

Response: According the suggestion of the reviewer, some keywords were deleted to avoid overlap the title.

  1. Immobilization of N fertilizer can occur rapidly in soil (over a few weeks) but this N is released slowly. Some authors reported that 7% of the 15N-labelled organic nitrogen was taken up by the first cereal crop after the labeled fertilizer had been applied, 4% by the second crop, 2% by the third and fourth crops, and 55% of the immobilized 15N remained in the soil at the end of 4 years. Please consider the issues related to the reduction of N plant bioavailability in the conditions of using straw without additional nitrogen fertilization (L. 41-44).

Response: We are grateful with the reviewer’s kind comments. We have referred the issues related to the reduction of N plant bioavailability in the conditions of using straw without additional nitrogen fertilization, and supplenments it in the introduction as follows. “The addition of wheat straw contributed to increased plant nitrogen (N) uptake through the alternation between N immobilization early in the season and N re-mineralization late in the season [2]. Whereas the N plant bioavailability may decease in the conditions of using straw without additional nitrogen fertilization as the rate of the organic nitrogen which can be taken up by the crop[3]. ”

  1. On what basis were the doses of mineral fertilization determined? (L.110-111, 113-114)

Response: The doses of mineral fertilization selected in the paper were based on the nutrient contents of soil for this experiment and the previous work in our research group (Wang et al, 2007). 

( Linxiao WANG, Al SHEN, Changlin KOU. Effects of organic fertilizer combined with nitrogen fertilizer on soil microbial biomass nitrogen and crop nitrogen utilization under wheat-maize rotation. Journal of Henan Agricultural Sciences, 2007,6,96-99.)

  1. The methodology does not mention a 10-year experiment, which is written in the abstract and the captions under the figures. 

Response: Thank the reviewer very much for the kind comment. “A 10-year field experiment was established since 2006 in Junxian County” has been added in the method.

  1. All methods used should be adequately described in the Materials and Methods section, e.g. method of available nitrogen determination. 

Response: Thank the reviewer for the kind comment. The methods have been added in the Materials and Methods section.

  1. Please correct editing errors, e.g. HClO4 (L. 158), significantdifferences (L. 259)

Response: We are very sorry for the editing errors . And these mistakes have been corrected in the revised paper.

  1. What was the basis of this statement: L. 464-467, 560-565.

Response: thanks reviewer for the kind comment. We have deleted the statements in L464-467, 560-565 as there was no direct evidence to support them based on the current results.

Reviewer 2 Report

In this manuscript, the authors analyzed soil microbial biomass and composition in a 10-year field experiment with different rates of straw return (50%, 100%) and N fertilizer (270, 360, 450, 540 kg Nha−1 yr−1) by phospholipid fatty acid analysis and high-throughput sequencing. The study objective is novel and findings obtained from this study will lead the scientific knowledge on the topic. However, I have some observations which should to be addressed by the authors.

1.       The N rates (450 and 500 kg/ha/yr) used in this experiment seem to be higher. I am not sure if it is for two crops (wheat and maize) together. If so, may be OK.

2.       Authors termed the control treatment as blank which is not appropriate. It is more scientific to denote it as a control rather than a blank.

3.       The PLFA in line 19 can be used only when it will be abbreviated in line 16 after ‘phospholipid fatty acid’.

4.       Lines 25-26, the genera of fungi should be italic.

5.       The term ‘dominate’ used in line 25 and 570 should be ‘dominant’.

6.       In line 122, please delete extra space after P2O5. ‘Potassium’ will be ‘potassium’.

7.       In line 522, the stop (.) needs to be deleted.

Author Response

  1. The N rates (450 and 500 kg/ha/yr) used in this experiment seem to be higher. I am not sure if it is for two crops (wheat and maize) together. If so, may be OK.

Response: The dosage of nitrogen fertilizer was for two crops together, and in the same year the dosage of nitrogen fertilizer for wheat and maize is equal to each other.

  1. Authors termed the control treatment as blank which is not appropriate. It is more scientific to denote it as a control rather than a blank.

Response: Thank the reviewer’s comment. The “(blank)” was deleted in th methods and the related discription about the control in paper and figures have been corrected .

  1. The PLFA in line 19 can be used only when it will be abbreviated in line 16 after ‘phospholipid fatty acid’.

Response: Thank the reviewer’s comment, we have added the abbreviation “PLFA” after ‘phospholipid fatty acid’ in line 16.

  1. Lines 25-26, the genera of fungi should be italic. The term ‘dominate’ used in line 25 and 570 should be ‘dominant’. In line 122, please delete extra space after P2O5. ‘Potassium’ will be ‘potassium’. In line 522, the stop (.) needs to be deleted.

Response: We are very sorry for errors in spelling and editing. And these mistakes have been corrected in the revised paper.

Reviewer 3 Report

The study is important to know that which ratio of the soil organic amendment as straw combined with fertilizer would be the most beneficial to maintain yield and ecosystem health.  

The urea is used as a N fertilizer in this study. Soils with the pH higher than 7 can result a significant ammonia volatilization as another loss of N applied from the fertilizer which should be accounted.

The assumption or claim that the 100% straw return into the soil would increase the risk of plant pathogen fungal distribution would have been visualised in plant health during the ten year experiment. Do you have information about the plant health status during the experiment? Did you use fungicides during the experiment?

The experimental design was a two-way split-plot, while as I see the evaluation was made by one-way ANOVA. Why do not use two-way ANOVA during evaluation of the results?

 L144:  Is the available nitrogen means the sum of nitrate+ammonium-N?

L507: Bacteroidetes instead of Bacteroides?

L509: Proteobacteria instead of Protobacteria.

Author Response

  1. The urea is used as a N fertilizer in this study. Soils with the pH higher than 7 can result a significant ammonia volatilization as another loss of N applied from the fertilizer which should be accounted.

Response: we agree with the reviewer. And a ammonia volatilization as another loss of N applied has been discussed in the revised paper.

  1. The assumption or claim that the 100% straw return into the soil would increase the risk of plant pathogen fungal distribution would have been visualised in plant health during the ten year experiment. Do you have information about the plant health status during the experiment? Did you use fungicides during the experiment?

Response: thank the reviewer for suggestion. Since the experiment was established in 2006 for the purpose of optimizing the rate of straw returning and nitrogen fertilizer input, it focused on the changes of soil nutrients and crop yield, and did not consider the relationship betweem the crop health and soil health for that time. Therefore, the use of pestcide and fungcide was carried out according to the actual situation of the current crop, which was not recorded strictly. It was an interesting discovery that the abundance of plant pathogen in soil had an increasing trend based on the results of this study, so we have planned to trace interrelation between the plant pathogen in soil and the health of plant in future.

  1. The experimental design was a two-way split-plot, while as I see the evaluation was made by one-way ANOVA. Why do not use two-way ANOVA during evaluation of the results?

Response: We are grateful with the reviewer’s kind comments. We have reanalysed the results by two-way ANOVA. The results showed that they were generally consistent with the previous analysis, but the two-factor analysis could reflect the overall influence of straw and nitrogen fertilizer on different treatments more directly. The results and figures were revised in red in the reveised paper.

  1. L144:  Is the available nitrogen means the sum of nitrate+ammonium-N?

Response: The available nitrogen was determined by the alkaline hydrolysis diffusion method. It contains nitrate-N, ammoonium-N and some low-molecular-weight hydrolyzable organic nitrogen. The method of available nitrogen determination have been added in the part of method.

  1. L507: Bacteroidetes instead of Bacteroides?L509: Proteobacteria instead of Protobacteria.

Response: We apologize for our negligence. “Bacteroidetes” and “Proteobacteria” have been instead of “Bacteroides” and “Protobacteri” in related lines in the revised paper.

Reviewer 4 Report

The manuscript titledResponse of soil environment and microbial community structure to different ratio of long-term straw return and nitrogen fertilizer in wheat-maize system provides information on the long-term effect of incorporating straw into the soil with different doses of nitrogen fertilizer on the physicochemical properties, heavy metal content and structure of the soil microbial population. In this sense, the data provided contribute to the knowledge of the long-term effects of straw incorporation into the soil and, therefore, the subject falls within the scope of the Sustainability journal, and might be of interest for its readership. Specific comment and suggestions to the authors are included in order to improve the final version of the manuscript.

 General comments:

 - The authors do not state their working hypotheses and expected results. The conclusions should answer these hypotheses or questions.

 - The authors should give details of the timing of soil sampling within the crop rotation, and the timing of soil sampling in relation to the time elapsed since the incorporation of straw into the soil. The latter may condition the result obtained on soil microbial composition.

 - Why is the concentration of heavy metals in the soil analyzed? Do any of the treatments (straw incorporation or nitrogen fertilizer doses) involve an input of heavy metals into the soil? How do you explain the differences in heavy metals in the soil? They do not seem to be entirely related to the treatments (it would be convenient to provide ANOVA results for heavy metal concentration as a function of straw incorporation and as a function of fertilizer treatment).

 - The authors relate high nitrogen availability associated with low risk of nitrate leaching, but this is not always the case. It will depend on whether there are plants present at that time that can extract that N; if there are no plants or they are at a small growth stage, the risk of nitrate leaching at high soil contents is very high. To speak of leaching risk, it would be better to know the available N content throughout the crop cycle (N dynamics in the soil); with only one soil sampling, it is daring to speak of leaching risk. It would also be convenient to know the N content in the form of nitrate.

 - On the other hand, the doses of N supplied are very high, so it is very likely that there will be significant losses due to nitrate leaching. These are not adequate doses for a wheat-maize rotation. 

 - The statistical analysis should be adequately explained. Was simple or factorial ANOVA done? What were the factors and their levels? In addition, the results of the ANOVA should be shown. Are there statistically significant differences for each of the physicochemical properties according to the dose of straw and N used?

 Specific remarks:

 - Line 19: Please, define PLFAs; for example, in line 16 after phospholipid fatty acid analysis (PLFAs)

- Line 22: Please, define F/B ratio

- Line 27: (> 450 kg ha-1 y-1 added)

- Line 99: from 532 to 1381 mm

- Line 100: from 1848 to 2489 h

- Line 122: potassium chloride

- Lines 136 – 138: Were three samplings performed or do you mean three analytical repetitions? If three samples were taken, on what dates were they taken?

- Line 146: Kjeldahl

- Line 158: HCLO4

- Line 192 and others: µL

- Line 229: …software (CANOCO…

- Line 435: fungi unclassified

- Lines 464 – 467: If more N is available, how is the risk of nitrate leaching reduced? It will depend, in any case, on whether that N is being taken up by the plant or not. If the highest available N content coincides when there is no crop or the crop is in its early stages of growth, then the risk of leaching losses increases.

- lines 471-473: microbial biomass C has not been measured. Is PLFAs analysis a measure of microbial biomass? Microbial biomass C has not been measured.

- Line 478: soil [33].

- Lines 491 – 493: But what is the value of the F:B ratio that indicates the stability of the soil ecosystem?

- Lines 514 – 517: And how do you explain the result for the S1N3 treatment?

Author Response

Dear Editor and Reviewers,

We highly appreciate the valuable comments of the referees on our manuscript of “sustainability-2082472”. The suggestions are quite helpful for us and we have incorporated them into the revised paper. During the last week, we have referred to literatures and papers and revised the paper to improve the quality. As below, on behalf of my co-authors, I would like to clarify some of the points raised by the reviewers. The revised sentences of the manuscript were presented in red font in the revised paper, please see the attachment. And we hope the reviewers and the editors will be satisfied with our responses to the ‘comments’ and the revisions for the original manuscript.

Thanks and best regards!

Yours sincerely,

Man Yu

2023-01-08

Reviewer #4

The manuscript titled “Response of soil environment and microbial community structure to different ratio of long-term straw return and nitrogen fertilizer in wheat-maize system provides information on the long-term effect of incorporating straw into the soil with different doses of nitrogen fertilizer on the physicochemical properties, heavy metal content and structure of the soil microbial population. In this sense, the data provided contribute to the knowledge of the long-term effects of straw incorporation into the soil and, therefore, the subject falls within the scope of the Sustainability journal, and might be of interest for its readership. Specific comment and suggestions to the authors are included in order to improve the final version of the manuscript.

General comments:

  1. The authors do not state their working hypotheses and expected results. The conclusions should answer these hypotheses or questions.

Response: We are grateful with the reviewer’s kind comments. And the conclusions were rewriten as follows: “The straw returning with N fertilizer incresed the contents of most soil nutrient, especially in teatment at 50% straw return with 450 kg N ha−1 yr−1, maily for higher SOC, TN, AK and AP contents. The structure of soil microbial communities were also significantly affected by straw return and N fertilizer. Total PLFAs indicated that straw return combined with N fertilizer promoted soil microbial growth and increased biomass. The F:B ratio showed that 100% straw with 450 kg N ha−1 yr−1 was not conducive to the stability of the soil ecosystem. According to the heatmap analysis, the similar rate of straw returning and the similar level of nitrogen fertilizer application will be divided into the same cluster. But some generas Gibberella, Sarocladium, Pseudallescheria, Mycosphaerella, and Microdochium which are either saprophytic fungi in soil or pathogens of plants became the dominate fungi generas in the treatments with 100% straw returning combining higher N fertilizer ( > 450 kg ha−1 yr−1) added. The fungal communities were significantly affected by heavy metals (Hg, Pb, and Cr), SOC, AN, and the C/N ratio. In addition, the relative abundances of some heavy metal-tolerant bacteria such as those in the Proteobacteria and Chlorobi increased in the soils in the 100% straw return treatments. Thus, the results of this study revealed that long-term 100% straw returning combining higher N fertilizer ( > 450 kg ha−1 yr−1) might lead to the risk of soil disease and pollution, while 50% straw returning combined with 450 kg ha−1 yr−1 might be more sustainable and beneficial to soil nutrient acccumulation and microbial community health in wheat-maize rotation fields.”

2 The authors should give details of the timing of soil sampling within the crop rotation, and the timing of soil sampling in relation to the time elapsed since the incorporation of straw into the soil. The latter may condition the result obtained on soil microbial composition.

Response: Thank the reviewer for his/her kind comments. The details of the timing of soil sampling within the crop rotation as follows: “Soil samples were collected 3 days after the wheat harvest prior to the straw returning and fertilizing on May 2016.”

3 Why is the concentration of heavy metals in the soil analyzed? Do any of the treatments (straw incorporation or nitrogen fertilizer doses) involve an input of heavy metals into the soil? How do you explain the differences in heavy metals in the soil? They do not seem to be entirely related to the treatments (it would be convenient to provide ANOVA results for heavy metal concentration as a function of straw incorporation and as a function of fertilizer treatment).

Response: Thank the reviewer for his/her kind comments. The concentration of heavy metals in the soil were analyzed to understanding whether a link exists or not between the increase in future soil N inputs and the environmental contaminants (e.g., heavy metals), as it is the potential threat to soil quality, agricultural production, human health, and the environment. In fact, they do not seem to be entirely related to the treatments in the present study reanalyed by two-way analysis of variance. However , the correlation betweem the heavy metal and some bacteria were positively in the 100% straw return treatments. Based on the quality of the results, we removed Figure 2 in the main text, and provide the results about heavy metal in soil in Figure S1 and Table S4 ( ANOVA results for heavy metal concentration as a function of straw incorporation and as a function of fertilizer treatment) in the “Supplementary data”.

4 The authors relate high nitrogen availability associated with low risk of nitrate leaching, but this is not always the case. It will depend on whether there are plants present at that time that can extract that N; if there are no plants or they are at a small growth stage, the risk of nitrate leaching at high soil contents is very high. To speak of leaching risk, it would be better to know the available N content throughout the crop cycle (N dynamics in the soil); with only one soil sampling, it is daring to speak of leaching risk. It would also be convenient to know the N content in the form of nitrate.

Response: We agree with the reviewer. It is not rigorous to relate high nitrogen availability associated with low risk of nitrate leaching and the statement was deleted in the revision. The available N content in soil was more closely correlated with the current soil, crop cycle and environmental parameters than soil TN. It would be better to know the available N content and N pollution to environment throughout the crop cycle with N dynamics by obseving the different forms of nitrogen in soil, water and atmosphere completely. Thanks for the suggestion of the reviewer, we are conducting relevant work, and hoping reveal the effect on enviornment of N fertilizer applied. Related discussion was rewrittern in the revised paper as follows: “In addition, the pH in soil in the treatments were higher than 7 which may result a significant ammonia volatilization as another loss of N[8]. Soil AN was more closely correlated with the current soil, plant and environmental parameters than soil TN. In the 50% and 100% straw return treatments, depending on the amount of fertilizer, soil AK and AP content also improved significantly, while the soil AN did not increse with the incease of N input. In the 50% straw return treatments with 450 kg N ha−1 yr−1, the TN was the highest with a lower AN, it indicated that the organic nitrogen in soil was higher in the S1N3.”

5 On the other hand, the doses of N supplied are very high, so it is very likely that there will be significant losses due to nitrate leaching. These are not adequate doses for a wheat-maize rotation. 

Response: The dosage of nitrogen fertilizer was for a wheat-maize rotation, and in the same year the dosage of nitrogen fertilizer for wheat and maize is equal to each other. The experimental N application rate followed the recommended N fertilizer application rate for sustaining crop yields and minimizing environmental costs in Chinese double cereal cropping systems. The doses of mineral fertilization selected in the paper were based on the nutrient contents of soil for this experiment and the previous work in our research group (Wang et al, 2007). 

( Linxiao WANG, Al SHEN, Changlin KOU. Effects of organic fertilizer combined with nitrogen fertilizer on soil microbial biomass nitrogen and crop nitrogen utilization under wheat-maize rotation. Journal of Henan Agricultural Sciences, 2007,6,96-99.)

 - The statistical analysis should be adequately explained. Was simple or factorial ANOVA done? What were the factors and their levels? In addition, the results of the ANOVA should be shown. Are there statistically significant differences for each of the physicochemical properties according to the dose of straw and N used?

Response: We are grateful with the reviewer’s kind comments. We have reanalysed the results by two-way ANOVA. The results showed that they were generally consistent with the previous analysis, but the two-factor analysis could reflect the overall influence of straw and nitrogen fertilizer on different treatments more directly. The statistically significant differences for each of the physicochemical properties according to the dose of straw and N used were provided in the “Supplementary data”. The results and figures were revised in red in the reveised paper.

  Specific remarks:

1 Line 19: Please, define PLFAs; for example, in line 16 after phospholipid fatty acid analysis (PLFAs);- Line 22: Please, define F/B ratio; - Line 27: (> 450 kg ha-1 y-1 added); Line 99: from 532 to 1381 mm; - Line 100: from 1848 to 2489 h; - Line 122: potassium chloride; - Line 146: Kjeldahl; - Line 158: HCLO4; - Line 192 and others: µL; - Line 229: …software (CANOCO…; - Line 435: fungi unclassified-; Line 478: soil [33].

Response: We are very sorry for the editing errors . And these mistakes have been corrected in the revised paper.

2 Lines 136 – 138: Were three samplings performed or do you mean three analytical repetitions? If three samples were taken, on what dates were they taken?

Response: We are very sorry for our unclear description. It means three analytical repetitions and related description has been corrected in the revised paper.

3 Lines 464 – 467: If more N is available, how is the risk of nitrate leaching reduced? It will depend, in any case, on whether that N is being taken up by the plant or not. If the highest available N content coincides when there is no crop or the crop is in its early stages of growth, then the risk of leaching losses increases.

Response: according to “General comments”, the statement in lines 464-467 was deleted as there was no direct evidence to support them based on the current results.

4 lines 471-473: microbial biomass C has not been measured. Is PLFAs analysis a measure of microbial biomass? Microbial biomass C has not been measured.

Response: thank the reviewer very much. Total PLFA has been suggested as a measure of the total microbial biomass, and it has been shown to correlate well with the biomass measurements using fumigation extraction technique in our previous work (Yu et al, 2009). In this experiment, the total PLFA was used to suggest the changes in microbial biomass. And the above sentences were added in related lines in the revision.

Yu M, Zeng GM. Influence of Phanerochaete chrysosporium on microbial communities and lignocellulose degradation during solid-state fermentation of rice straw. Proc Biochem 2009, 44,17-22

Again, we thank the editor(s) and reviewer(s) very much for your hard work on the paper. Wish you have a wonderful Day.

Round 2

Reviewer 4 Report

Thank you for the answers to the questions and doubts raised.